# Learning Early Treatment Strategy from Snapshots for mRNA-protein Regulatory Networks

## Abstract

Early disease detection with snapshot data has been effectively addressed by the Dynamical Network Biomarkers (DNBs) theory. After early disease detection, it is crucial to consider early medical treatment to prevent it. This paper presents a novel framework for identifying mRNA-protein regulatory systems from snapshot data and designing interventions. We first estimate the state covariance of mRNA-protein expression using multi-episode snapshot samples. Then, we identify the underlying continuous-time dynamics by solving a Lyapunov-based regression problem. We provide finite-sample guarantees on the estimation accuracy of the system matrix and its dominant eigenvectors, which are essential for downstream treatment design. Building on these estimates, we formulate an optimal re-stabilization strategy that minimizes input energy with desired spectral shifts. To ensure practical feasibility, we further propose a diagonal re-stabilization scheme that identifies key regulatory nodes using a first-order eigenvalue sensitivity analysis. Numerical examples on synthetic mRNA-protein network demonstrate that our method accurately identifies regulatory node under high-dimensional, low-sample conditions and significantly outperforms existing baselines.

## 1 Introduction

In biological systems, functions such as mRNA regulation and protein interactions occur through complex networks (Briat & et al., 2016). Many diseases stem from abrupt deterioration in these networks, often modeled as bifurcation phenomena (Chen et al., 2012; Sadria & Bury, 2024). Using the snapshot data samples, the *Dynamical Network Biomarker* (DNB) method can predict the stage of the system immediately before the bifurcation occurs, referred to here as the pre-disease stage (Liu et al., 2015). According to DNB theory, certain nodes exhibit amplified fluctuations as they approach a bifurcation point. These dominant directions reflect nodes with large fluctuations, and can be estimated from snapshot data. The DNB theory also enables early disease detection based on the increased fluctuations of specific biomarkers (Aihara et al., 2022). The ability to detect diseases at the pre-disease stage is crucial for early medical intervention. Traditional Japanese medicine has been used to suppress DNB node fluctuations and prevent disease progression (Koizumi et al., 2020). Experimental results in (Chen & et al., 2022) demonstrate that manipulating multiple DNB nodes can significantly alter malignant phenotypes in lung cancer. The success of heuristic approaches has further inspired the development of theoretical frameworks for early medical intervention. Recent studies have explored early treatment via high-dimensional low sample-size (HDLSS) snapshot data. While (Yasukata et al., 2023) proposed a single-input method for undirected networks, extensions to directed networks were developed in (Shen et al., doi:10.1109/TETCI.2024.3442824). These works highlight the importance of the system matrix's left eigenvector corresponding to eigenvalues with maximal real parts for optimal input placement, information not directly accessible via principal component analysis (PCA), necessitating system identification by snapshot data. Learning stochastic dynamics from snapshot data has recently gained considerable attention in the machine learning community (Song et al., 2021; Neklyudov et al., 2023; Tong et al., 2024). A dominant approach involves first inferring time-series trajectories from the data, followed by system identification based on the inferred trajectories (Tong et al., 2020). This trajectory inference has been particularly studied in the context of single-cell RNA sequencing (Saelens et al., 2019; Shi et al., 2022; Sha et al.,

2024), where uncovering the underlying temporal progression of cellular states is essential. Optimal transport methods have become central tools for such snapshot datasets with temporal resolution (Bunne et al., 2024; Schiebinger et al., 2019), with the Schrödinger bridge (SB) formulation extending these methods to stochastic dynamics by modeling the most likely stochastic paths between two distributions relative to a reference process (Léonard, 2014; Shi et al., 2024; Liu et al., 2022). To improve computational tractability, regularization techniques have also been introduced to these transport-based formulations (Chen et al., 2022; Zhang et al., 2025). Despite these advances, such methods often suffer from high computational cost and rely on large sample sizes to obtain reliable results, which are rarely satisfied in single-cell data analysis.

A core challenge in modelling gene regulatory systems is their inherently complex multiscale character: slow transcriptional changes interact with much faster protein-level dynamics, and meaningful mechanistic descriptions and models must account for different scales (Fletcher & Osborne, 2022). Classical mechanistic models based on differential equations have been used in systems biology for capturing these interactions across domains from circadian–metabolic coupling and metabolic regulation to mechanistic models of immune and viral dynamics (Sadria & Layton, 2021a;b; Ingalls, 2013). While these models have enabled mechanistic insight across many biological settings, they typically require dense time-series or carefully designed experiments for reliable parameter identification, which are scarce in current high-throughput single-cell genomics. These limitations have motivated powerful trajectory-inference and representation-learning methods that reconstruct temporal progressions from snapshots (for example, transport-based formulations and Schrödinger-bridge approaches). Such tools are useful for recovering likely cellular paths and population-level flows, but their design goals differ from those needed for control: they prioritize path reconstruction (often under specific loss/regularization choices) rather than explicit recovery of the low-level regulatory parameters and multiscale structure required to reason about stability or to design energy-efficient interventions. Complementary recent work demonstrates scalable deep and representation-learning approaches that predict fate changes or extract parsimonious dynamical models from single-cell data and a library-guided sparse discovery framework (Sadria et al., 2022; Sadria & Swaroop, 2025); these advances substantially improve fate prediction and model discovery, but by themselves do not directly yield the identifiable, control-ready parameterizations we target here.

In contrast, in this paper, we leverage the structural properties of the mRNA–protein regulatory network to achieve reliable and computationally efficient system identification. This identified system enables the design of effective early treatment strategies. Our main contributions are summarized as follows: (a) **System identification from snapshot data:** We develop a framework that uses structural constraints of the mRNA–protein regulatory network to identify the system matrix from finite snapshot data via Lyapunov-based regression. (b) **Theoretical guarantees:** We establish finite-sample confidence bounds for both the system matrix estimation and the associated eigenvectors, ensuring reliability even under High-Dimension Low-Sample-Size (HDLSS) conditions. (c) **Early intervention design:** Building on the estimated system, we design both optimal and diagonal re-stabilization strategies, providing a practical approach for early treatment at the pre-disease stage. This is crucial as early (pre-disease) interventions are significantly more effective and less invasive than treatments applied after full disease onset.

## 2 PRELIMINARIES AND BACKGROUNDS

**Dynamic system for gene regulation.** Gene transcription is regulated by transcription factors that bind to DNA, with transcription rates modulated by their concentrations. Translated proteins can further regulate gene expression. Translation lacks feedback to mRNA, and both mRNAs and proteins degrade stochastically. This gene-mRNA-protein feedback is modeled by (Chen et al., 1999; Liu et al., 2016; Passemiers et al., 2022; Sanders et al., 2020; Weidmann et al., 2021):

$$\dot{\mathbf{z}} = \mathbf{F}_{\boldsymbol{\theta}}\left(\mathbf{z}\right) + \mathbf{w}, \quad \mathbf{F}_{\boldsymbol{\theta}}\left(\mathbf{z}\right) = \begin{bmatrix} -\Delta_{\mathsf{r},\boldsymbol{\theta}}\mathbf{z}_{\mathsf{r}} + \mathbf{f}_{\boldsymbol{\theta}}(\mathbf{z}_{\mathsf{p}}) \\ \Gamma_{\mathsf{r},\boldsymbol{\theta}}\mathbf{z}_{\mathsf{r}} - \Delta_{\mathsf{p},\boldsymbol{\theta}}\mathbf{z}_{\mathsf{p}} \end{bmatrix}. \tag{1}$$

where $\mathbf{z} := [\mathbf{z}_{\mathsf{r}}, \mathbf{z}_{\mathsf{p}}]^{\top} \in \mathbb{R}^{2n}$ denotes mRNA and protein concentrations, and $\mathbf{w}$ is Gaussian noise with covariance $\mathbf{D}$. Transcription function $\mathbf{f}_{\boldsymbol{\theta}}(\mathbf{z}_{\mathsf{p}})$ is a nonlinear $n$-dimensional vector encoding protein-mediated regulation. Diagonal matrices $\Gamma_{\mathsf{r},\boldsymbol{\theta}}$, $\Delta_{\mathsf{r},\boldsymbol{\theta}}$, and $\Delta_{\mathsf{p},\boldsymbol{\theta}}$ represent translation and degradation rates and are all non-degenerate. These quantities are parameterized by $\boldsymbol{\theta} \in \Theta \subset \mathbb{R}^{m}$. Let $\mathbf{z}^{\mathsf{e}}$ be the equilibrium point such that $\mathbf{F}_{\boldsymbol{\theta}}\left(\mathbf{z}\right) = 0$, where $\mathbf{z}^{\mathsf{e}} = [\mathbf{z}_{\mathsf{r}}^{\mathsf{e}}, \mathbf{z}_{\mathsf{p}}^{\mathsf{e}}]^{\top}$. A linearized approximation

around $\mathbf{z}^{\mathrm{e}}$ can be written by (Chen et al., 1999)

$$\dot{\mathbf{x}} = \mathbf{A}_{\boldsymbol{\theta}} \mathbf{x} + \mathbf{w}, \ \mathbf{x} = \mathbf{z} - \mathbf{z}^{\mathrm{e}}, \ \mathbf{A}_{\boldsymbol{\theta}} = \begin{bmatrix} -\Delta_{\mathrm{r},\boldsymbol{\theta}} & \Gamma_{\mathrm{p},\boldsymbol{\theta}} \\ \Gamma_{\mathrm{r},\boldsymbol{\theta}} & -\Delta_{\mathrm{p},\boldsymbol{\theta}} \end{bmatrix}, \ \Gamma_{\mathrm{p},\boldsymbol{\theta}} = \left. \frac{\partial \mathbf{f}_{\boldsymbol{\theta}}(\mathbf{z}_{\mathrm{p}})}{\partial \mathbf{z}_{\mathrm{p}}} \right|_{\mathbf{z}=\mathbf{z}^{\mathrm{e}}}. \tag{2}$$

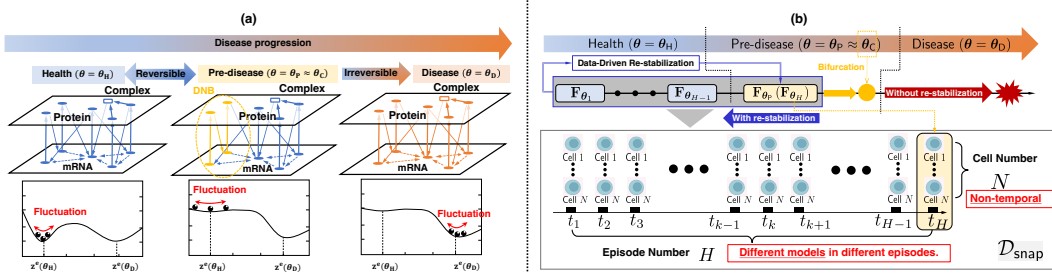

Figure 1: Conceptual illustration of the disease progression and snapshot dataset: (a) Illustration of the disease progression from the health stage to the disease stage through the pre-disease stage. (b) Illustration of the snapshot data-driven re-stabilization and the issue for system identification.

**Disease progress and snapshot data.** Figure 1 (a) shows chronic disease progression driven by the parameter $\boldsymbol{\theta}$, where a tipping point $\boldsymbol{\theta}_{\mathrm{C}}$ separates the healthy stage $\Theta_{\mathrm{H}}$ from the disease stage $\Theta_{\mathrm{D}}$. This transition corresponds to a bifurcation in the nonlinear system (Chen et al., 2012). Let $\overline{\Theta}_{\mathrm{H}} := \Theta_{\mathrm{H}} \cup \partial\Theta_{\mathrm{H}}$ denote the closure of the healthy region. As $\boldsymbol{\theta}$ evolves within $\Theta := \overline{\Theta}_{\mathrm{H}} \cup \Theta_{\mathrm{D}}$, the equilibrium $\mathbf{z}_{\boldsymbol{\theta}}^{\mathrm{e}}$ changes accordingly. In $\overline{\Theta}_{\mathrm{H}}$, the equilibrium is stable and approximated by $\mathbf{z}^{\mathrm{e,H}}$, while in $\Theta_{\mathrm{D}}$, it shifts to $\mathbf{z}^{\mathrm{e,D}}$, far from the healthy state (bottom of Figure 1 (a)). As shown in Figure 1 (a), the system is stable in both $\Theta_{\mathrm{H}}$ and $\Theta_{\mathrm{D}}$, since the maximum eigenvalue $\lambda_{\mathrm{d},\boldsymbol{\theta}}$ of $\mathbf{A}_{\boldsymbol{\theta}}$ has a significantly negative real part. At the tipping point $\boldsymbol{\theta} \in \partial\Theta_{\mathrm{H}}$, this eigenvalue becomes zero. In the pre-disease stage $\boldsymbol{\theta}_{\mathrm{P}} \approx \boldsymbol{\theta}_{\mathrm{C}}$, we have the real part $\mathrm{Re}\{\lambda_{\mathrm{d},\boldsymbol{\theta}}\} \approx 0^{-}$, indicating low stability and high sensitivity to perturbations. As illustrated in Figure 1(b), measurements of the system state $\mathbf{x}$ (mRNA and protein levels) are collected over $H$ episodes. Each episode $k$ corresponds to a biological sampling time $t_k$, where $N$ individual cells are measured. These single-cell observations form a snapshot of the internal state: $\mathcal{D}_{\mathrm{snap}}^{(k)} := \left\{\mathbf{x}_m^{(k)}\right\}_{m=1}^{N}, \quad \mathbf{x}_m^{(k)} \in \mathbb{R}^{2n}$. Such data are not time-series but population-level samples reflecting heterogeneity at $t_k$. The full dataset is denoted by $\mathcal{D}_{\mathrm{snap}} := \left\{\mathcal{D}_{\mathrm{snap}}^{(k)}\right\}_{k=1}^{H}$.

**Assumption 1.** *For each $k = 1, \ldots, H$, the following holds: (1) The samples $\left\{\mathbf{x}_m^{(k)}\right\}_{m=1}^{N}$ are i.i.d. from a continuous distribution $\mu_k$: $\mathbf{x}_m^{(k)} \sim \mu_k$. (2) $\boldsymbol{\theta}$ evolves smoothly across time $t_k$[1].*

This setting reflects realistic biological measurement conditions, where mRNA-protein regulatory systems are quasi-stationary during sampling, and population-level expression distributions shift gradually due to slow parameter changes. Note that the i.i.d. assumption applies only to the sample index $m$, and no independence across time step $k$ is assumed.

## 3 ADDRESSED PROBLEM AND CHALLENGING ISSUES

Pre-disease stage can be efficiently detected by snapshot data. Details of pre-disease detection are summarized in Appendix A. Once the pre-disease stage is detected, it is natural to consider medical interventions aimed at preventing further progression and restoring the gene regulatory network to the healthy stage. In addition to alleviating patient suffering, early intervention at the pre-disease stage is generally more effective—and often less invasive—than treating fully developed diseases. Recovery from the disease stage requires steering the system from a diseased equilibrium point $\mathbf{z}_{\boldsymbol{\theta}}^{\mathrm{e}}$ with $\boldsymbol{\theta} \in \Theta_{\mathrm{D}}$ back to a healthy equilibrium $\mathbf{z}_{\boldsymbol{\theta}}^{\mathrm{e}}$ with $\boldsymbol{\theta} \in \Theta_{\mathrm{H}}$. This constitutes a nonlinear control

---

[1]"Smooth" means that $\boldsymbol{\theta}$ is a smooth function of time $t$, indicating a smooth parameter evolution in time.

problem due to the large deviation between these equilibria and the complex dynamics involved. In contrast, recovery from the pre-disease stage is more tractable. Since the equilibrium point $\mathbf{z}_{\boldsymbol{\theta}}^{\mathrm{e}}$ changes only slightly for $\boldsymbol{\theta} \in \overline{\Theta}_{\mathsf{H}}$, the problem can be reasonably approximated as a linear control task. The objective in this case is not to shift the equilibrium but to enhance the system's robustness to external perturbations by increasing its local stability margin. To distinguish this type of intervention from full recovery, we refer to it as *re-stabilization*. Re-stabilization aims to shift the dominant eigenvalue $\lambda_{\mathsf{d},\boldsymbol{\theta}}$ further into the left-half complex plane, thereby enhancing the system's resilience while it remains in the vicinity of the healthy regime. This can be achieved by introducing a feedback loop that modifies the system matrix to $\mathbf{A}_{\boldsymbol{\theta}} + \mathbf{B}_{\boldsymbol{\theta}}\mathbf{K}_{\boldsymbol{\theta}}$. Here, the matrix $\mathbf{B}_{\boldsymbol{\theta}} = [\mathbf{b}_{1,\boldsymbol{\theta}}, \ldots, \mathbf{b}_{l,\boldsymbol{\theta}}] \in \mathbb{R}^{2n \times l}$ determines the input placement—i.e., the selection of genes for intervention. We now formulate the re-stabilization problem with snapshot dataset $\mathcal{D}_{\mathsf{snap}}$:

**Problem 1** (Re-stabilization problem)**.** *At each episode $k = 1, ..., H$, the parameter in the transcription function is defined by $\boldsymbol{\theta}$. Note that the true system matrix $\mathbf{A}_{\boldsymbol{\theta}_k}$, $k = 1, ..., H$ in the linearized error dynamics equation 2 is unknown. Given a snapshot dataset $\mathcal{D}_{\mathsf{snap}}$, the objective is to design a feedback intervention $\mathbf{B}_{\boldsymbol{\theta}_H}\mathbf{K}_{\boldsymbol{\theta}_H}$ that the dominant eigenvalue of the closed-loop system $\tilde{\lambda}_{\mathsf{d},\boldsymbol{\theta}} := \max \mathrm{Re}\left(\mathrm{eig}\left(\mathbf{A}_{\boldsymbol{\theta}} + \mathbf{B}_{\boldsymbol{\theta}}\mathbf{K}_{\boldsymbol{\theta}}\right)\right)$ satisfies $\mathrm{Re}(\tilde{\lambda}_{\mathsf{d},\boldsymbol{\theta}}) < \mathrm{Re}(\lambda_{\mathsf{d},\boldsymbol{\theta}})$, realizing re-stabilization.*

By solving the re-stabilization problem, we increase the stability margin of the mRNA-protein regulatory system, making it less sensitive to random fluctuations. As a result, the state distribution becomes more concentrated around the healthy equilibrium and less likely to drift toward the disease state under uncertain perturbations. In biological terms, this means that even when the system is close to a critical transition, effective re-stabilization can suppress large fluctuations in regulatory genes and reduce the probability of crossing into the disease stage. This highlights the practical motivation of our framework: identifying regulatory nodes not only enables early detection of disease, but also provides actionable targets for early intervention.

**Challenging issues.** Solving Problem 1 faces several key challenges. First, the snapshot dataset $\mathcal{D}_{\mathsf{snap}}$ collected at the pre-disease stage ($k = H$), $\mathcal{D}_{\mathsf{snap}}^{(H)}$, suffers from the high-dimensional, low-sample-size (HDLSS) regime, as it typically contains expression measurements for $n > 10^4$ genes across a few thousands single cells at most. That is, the number of molecular features far exceeds the number of observable cells, creating severe challenges for statistical inference on the system matrix $\mathbf{A}_{\boldsymbol{\theta}_H}$ using only the available data at $t_H$. Second, the dataset $\mathcal{D}_{\mathsf{snap}}$ consists of non-temporal (snapshot) observations, which precludes the use of system identification techniques that rely on time-series trajectories. Furthermore, although snapshot data from earlier episodes are available, they originate from distinct underlying models due to variation in $\boldsymbol{\theta}$ across time and therefore cannot be directly pooled with data from the pre-disease stage. As a result, classical control techniques such as pole placement, which require full knowledge of the system matrix, are not applicable. These issues necessitate the development of a novel data-driven approach capable of designing re-stabilizing feedback using only distributional information extracted from limited snapshot data.

# 4 Proposed method: generative re-stabilization

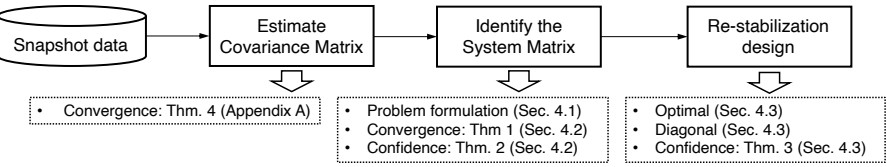

Figure 2: A brief summary of the proposed method and content in this section.

A summary of the proposed method is illustrated by Figure 2. We begin by estimating the covariance matrix of the state at the pre-disease stage using snapshot data. Based on the estimated covariance, the system matrix is then identified by solving a regression problem derived from the Lyapunov equation. We analyze the probabilistic convergence of the estimated system matrix and provide a finite-sample confidence bound for its estimation error. Finally, leveraging the estimated system matrix, we carry out re-stabilization design, including both the optimal re-stabilization and a practically implementable approximate diagonal re-stabilization approach.

## 4.1 Regression from Lyapunov equation

At the steady state, the covariance matrix of $\mathbf{x}$, denoted by $\mathbf{C}_{\boldsymbol{\theta}} \in \mathbb{R}^{2n \times 2n}$, satisfies the Lyapunov equation as follows (Chen et al., 2012; Oku & Aihara, 2018):

$$\mathbf{A}_{\boldsymbol{\theta}} \mathbf{C}_{\boldsymbol{\theta}} + \mathbf{C}_{\boldsymbol{\theta}} \mathbf{A}_{\boldsymbol{\theta}}^{\top} + \mathbf{D} = 0. \tag{3}$$

We identify the system matrix using equation 3 rather than solving a standard time-series regression problem. The motivation for introducing the Lyapunov equation-based formulation is that, in practice, we do not observe full time-series trajectories but only snapshot measurements across cells. The Lyapunov equation directly links the covariance of the stationary distribution to the system matrix, allowing system identification from non-temporal snapshot data. This formulation addresses the key challenge of snapshot-based identification by transforming it into a regression problem on covariance matrices, which can be consistently estimated from single-cell data. On the other hand, in our case, the non-zero elements of $\mathbf{A}_{\boldsymbol{\theta}}$ are known because $\Delta_{\mathsf{r},\boldsymbol{\theta}}$, $\Delta_{\mathsf{p},\boldsymbol{\theta}}$, $\Gamma_{\mathsf{r},\boldsymbol{\theta}}$ are diagonal matrices. Let $\mathscr{A}_{\boldsymbol{\theta}}, \mathscr{C}_{\boldsymbol{\theta}}, \mathscr{D}$ be defined as:

$$\mathscr{A}_{\boldsymbol{\theta}} = \mathsf{vec}(\mathbf{A}_{\boldsymbol{\theta}}), \ \ \mathscr{C}_{\boldsymbol{\theta}} = \mathbf{C}_{\boldsymbol{\theta}} \otimes \mathbf{I} + (\mathbf{I} \otimes \mathbf{C}_{\boldsymbol{\theta}})\mathbf{T}, \ \ \mathscr{D} = 2\mathsf{vec}(\mathbf{D}), \tag{4}$$

where $\mathbf{I}$ is the $2n \times 2n$ identity matrix and $\mathbf{T}$ is a transformation matrix satisfying $\mathsf{vec}(\mathbf{X}) = \mathbf{T}\mathsf{vec}(\mathbf{X}^{\top})$. From the Lyapunov equation equation 3, we obtain the linear equation:

$$\mathscr{C}_{\boldsymbol{\theta}} \mathscr{A}_{\boldsymbol{\theta}} = -\mathscr{D}. \tag{5}$$

Since both $\mathbf{C}_{\boldsymbol{\theta}}$ and $\mathbf{D}$ are symmetric, the linear system equation 5 has rank $n(2n + 1)$ at most. If we were to estimate all $4n^2$ variables in $\mathscr{A}_{\boldsymbol{\theta}}$, the system equation 5 would remain underdetermined. However, we know that the diagonal matrices $\Delta_{\mathsf{r},\boldsymbol{\theta}}$, $\Delta_{\mathsf{p},\boldsymbol{\theta}}$, $\Gamma_{\mathsf{r},\boldsymbol{\theta}}$ contain at least $3n^2 - 3n$ zero elements. This implies that there are $3n^2 - 3n$ index pairs $(i, j)$ for which $\mathbf{A}_{\boldsymbol{\theta}}(ij) = 0$. Define $\mathbf{u}^{(ij)} \in \mathbb{R}^{1 \times 4n^2}$ as a row vector such that $\mathbf{u}_k^{(ij)} = 1$ for $k = 2n(i - 1) + j$, and $\mathbf{u}_k^{(ij)} = 0$ otherwise. Then, $\mathbf{u}^{(ij)} \mathscr{A}_{\boldsymbol{\theta}} = 0$ imposes the constraint $\mathbf{A}_{\boldsymbol{\theta}}(ij) = 0$. In our case, these $3n^2 - 3n$ row vectors together form the constraint matrix $\mathbf{U} \in \mathbb{R}^{(3n^2 - 3n) \times 4n^2}$, leading to:

$$\mathbf{U}\mathscr{A}_{\boldsymbol{\theta}} = \mathbf{0}. \tag{6}$$

By incorporating the constraint equation 6 into equation 5, the extended linear system becomes:

$$\mathscr{C}_{\boldsymbol{\theta},\mathsf{ext}} \mathscr{A}_{\boldsymbol{\theta}} = \mathscr{D}_{\mathsf{ext}}, \quad \mathscr{C}_{\boldsymbol{\theta},\mathsf{ext}} = \begin{bmatrix} \mathscr{C}_{\boldsymbol{\theta}} \\ \mathbf{U} \end{bmatrix}, \quad \mathscr{D}_{\mathsf{ext}} = \begin{bmatrix} -\mathscr{D} \\ \mathbf{0} \end{bmatrix}. \tag{7}$$

The augmented coefficient matrix has rank at most $n(2n + 1) + 3n^2 - 3n = 5n^2 - 2n$, which exceeds $4n^2$ for $n > 2$. This augmentation allows the unique determination of the matrix $\mathbf{A}_{\boldsymbol{\theta}}$.

The optimization problem for system identification, using $\mathscr{C}_{\boldsymbol{\theta},\mathsf{ext}}$ transformed from the real covariance matrix $\mathbf{C}_{\boldsymbol{\theta}}$, can be formulated as:

$$\min_{\mathscr{A}} J(\mathscr{A}, \mathbf{C}_{\boldsymbol{\theta}}) := \|\mathscr{C}_{\boldsymbol{\theta},\mathsf{ext}} \mathscr{A} - \mathscr{D}_{\mathsf{ext}}\|_2^2. \tag{$\mathcal{P}_{\boldsymbol{\theta}}$}$$

The solution of the optimization problem written by equation $\mathcal{P}_{\boldsymbol{\theta}}$ is unique as $\mathscr{A}_{\boldsymbol{\theta}}$ satisfying $J(\mathscr{A}_{\boldsymbol{\theta}}, \mathbf{C}_{\boldsymbol{\theta}}) = 0$ corresponds exactly to the true system matrix $\mathbf{A}_{\boldsymbol{\theta}}$.

## 4.2 Approximate regression problem

Although solving the optimization problem written by equation $\mathcal{P}_{\boldsymbol{\theta}}$ yields the true system matrix $\mathbf{A}_{\boldsymbol{\theta}}$, it requires knowledge of the covariance matrix $\mathbf{C}_{\boldsymbol{\theta}}$, which is unknown in practice. Let $\widehat{\mathbf{C}}_{\boldsymbol{\theta}}$ be an estimate of the true covariance matrix $\mathbf{C}_{\boldsymbol{\theta}}$ and formulate the following approximate problem:

$$\min_{\mathscr{A}} J\left(\mathscr{A}, \widehat{\mathbf{C}}_{\boldsymbol{\theta}}\right) := \|\widehat{\mathscr{C}}_{\boldsymbol{\theta},\mathsf{ext}} \mathscr{A} - \mathscr{D}_{\mathsf{ext}}\|_2^2. \tag{$\widehat{\mathcal{P}}_{\boldsymbol{\theta}}(\mathcal{D}_{\mathsf{snap}})$}$$

Here, $\widehat{\mathscr{C}}_{\boldsymbol{\theta},\mathsf{ext}}$ is the matrix transformed from $\widehat{\mathbf{C}}_{\boldsymbol{\theta}}$ by equation 4. Let $\widehat{\mathscr{A}_{\boldsymbol{\theta}}}$ denote the optimal solution to Problem $\widehat{\mathcal{P}}_{\boldsymbol{\theta}}(\mathcal{D}_{\mathsf{snap}})$, and let $\widehat{\mathbf{A}}_{\boldsymbol{\theta},\mathsf{est}}$ denote the final estimate of the true system matrix $\mathbf{A}_{\boldsymbol{\theta}}$, transformed from $\widehat{\mathscr{A}_{\boldsymbol{\theta}}}$. Since the optimization problems defined by equation $\widehat{\mathcal{P}}_{\boldsymbol{\theta}}(\mathcal{D}_{\mathsf{snap}})$ and equation $\mathcal{P}_{\boldsymbol{\theta}}$

differ due to the discrepancy between $\widehat{\mathscr{C}}_{\boldsymbol{\theta},\text{ext}}$ and $\mathscr{C}_{\boldsymbol{\theta},\text{ext}}$, it is necessary to examine whether the estimated matrix $\widehat{\mathbf{A}}_{\boldsymbol{\theta}}$ converges to the true system matrix $\mathbf{A}_{\boldsymbol{\theta}}$. In the rest part of this subsection, we will introduce the method of estimating $\widehat{\mathbf{C}}_{\boldsymbol{\theta}}$ by kernel conditional density estimator and then give the convergence analysis for $\widehat{\mathscr{A}_{\boldsymbol{\theta}}}$. By employing a kernel conditional density estimator to generate additional samples, we alleviate the challenge posed by the HDLSS regime.

**Kernel conditional density estimation.** Nadaraya–Watson (NW) conditional density estimator (Gooijer & Zerom, 2003; Hall et al., 1999) is used to approximate the conditional density (CDE) $p^{\mathsf{c}}_{\mathbf{X}}(\mathbf{x} \mid t)$. The first step is to estimate the joint probability density $p(\mathbf{x}, t)$ from the snapshot dataset $\mathcal{D}_{\mathsf{snap}}$ using kernel density estimation (KDE). Let $\widehat{p}(\mathbf{x}, t)$ denote the KDE computed from $\mathcal{D}_{\mathsf{snap}}$, defined as $\widehat{p}(\mathbf{x}, t) = \frac{1}{NH \cdot h} \sum_{k,m} K_{\mathbf{X}} \left( \frac{\mathbf{x} - \mathbf{x}_m^{(k)}}{h} \right) K_T \left( \frac{t - t_k}{h} \right)$, where $K_{\mathbf{X}}(\cdot)$ and $K_T(\cdot)$ are kernel functions for $\mathbf{X}$ and $t$, respectively, and $h$ is a smoothing parameter known as the bandwidth. Here, bandwidth $h$ satisfying the standard consistency conditions (as in (Gooijer & Zerom, 2003)): $h \to 0$ and $NHh^n \to \infty$ as $N, H \to \infty$. Various kernel functions can be used in practice, including uniform, triangular, biweight, triweight, Epanechnikov (parabolic), normal, among others. The NW conditional density estimator can then be computed as $\widehat{p}^{\mathsf{c}}_{\mathbf{X}}(\mathbf{x} \mid t) = \left\{ \sum_{k,m} K_{\mathbf{X}} \left( \frac{\mathbf{x} - \mathbf{x}_m^{(k)}}{h} \right) K_T \left( \frac{t - t_k}{h} \right) \right\} / \left\{ \sum_{k,m} K_T \left( \frac{t - t_k}{h} \right) \right\}$. It is important to note that each $\mathbf{x}_m^{(k)}$ is associated with the corresponding time point $t_k$. Thus, there are effectively $N$ samples of $\mathbf{x}$ corresponding to each $t_k$, even though $t_k$ itself appears only once per episode. Note that the true conditional density $p^{\mathsf{c}}_{\mathbf{X}}(\mathbf{x} \mid t_k)$ is Gaussian $\mathcal{N}(\mathbf{0}^n, \mathbf{C}_{\boldsymbol{\theta}})$ with zero mean and finite covariance. Zero-mean Gaussian setting is reasonable, since the system equation 2 is linearized at the equilibrium point and the measurements are collected at steady state, where the system is driven by white Gaussian noise.

**Covariance matrix estimation.** Let $\widehat{p}^{\mathsf{c}}_{\mathbf{X}}(\mathbf{x} \mid t_k)$ be the NW CDE constructed from the snapshot dataset $\mathcal{D}_{\mathsf{snap}}$. Let $\{\widehat{\mathbf{x}}_m^{(k)}\}_{m=1}^M$ be a set of i.i.d. samples generated from $\widehat{p}^{\mathsf{c}}_{\mathbf{X}}(\mathbf{x} \mid t_k)$. Define the sample covariance matrix associated $\widehat{p}^{\mathsf{c}}_{\mathbf{X}}(\mathbf{x} \mid t_k)$ by $\widehat{\mathbf{C}}_{\boldsymbol{\theta}} := \frac{1}{N} \sum_{m=1}^N \widehat{\mathbf{x}}_m^{(k)} \left( \widehat{\mathbf{x}}_m^{(k)} \right)^\top$.

Then, we have the following theorem regarding the convergence of $\widehat{\mathbf{A}}_{\boldsymbol{\theta}}$ to $\mathbf{A}_{\boldsymbol{\theta}}$.

**Theorem 1.** *As $N, H \to \infty$, we have $\widehat{\mathbf{A}}_{\boldsymbol{\theta}} \xrightarrow{w.p.1} \mathbf{A}_{\boldsymbol{\theta}}$.*

The proof of Theorem 1 is summarized in Appendix C. Furthermore, we investigate the confidence level of the estimation $\widehat{\mathbf{A}}_{\boldsymbol{\theta}}$ when the sample size of the snapshot dataset $\mathcal{D}_{\mathsf{snap}}$ is finite. To facilitate the analysis, we introduce the following mild assumption, which ensures the boundedness of the sample moments and is commonly adopted in finite-sample analyses.

**Assumption 2.** *There exists a constant $L_\beta > 0$ such that, with probability at least $1 - \beta$, the sample $\mathbf{x}$ drawn from $\widehat{p}^{\mathsf{c}}_{\mathbf{X}}(\mathbf{x} \mid t_k)$ satisfies $\|\mathbf{x}\|_\infty \leq L_\beta$.*

Assumption 2 is not restrictive in practice, as it is typically satisfied when the support of the estimated density $\widehat{p}^{\mathsf{c}}_{\mathbf{X}}$ is bounded or sufficiently concentrated around its mode. It provides a high-probability guarantee for the boundedness of the generated samples, which facilitates the establishment of finite-sample confidence bounds in subsequent analysis.

The objective function $J(\mathscr{A}, \mathbf{C}_{\boldsymbol{\theta}})$ is strongly convex with respect to $\mathscr{A}$ since its Hessian satisfies $\nabla^2 J = 2\mathbf{M}^\top \mathbf{M}$ with $\mathbf{M} = \mathbf{C}_{\boldsymbol{\theta}}^\top \otimes \boldsymbol{I} + \boldsymbol{I} \otimes \mathbf{C}_{\boldsymbol{\theta}}$. Since $\mathbf{C}_{\boldsymbol{\theta}} \succ 0$, $\mathbf{M}$ is invertible, and thus $\mu := 2\lambda_{\min}\left(\mathbf{M}^\top \mathbf{M}\right) > 0$ gives the strong convexity modulus. Then, we give the following theorem regarding the confidence level of the estimation $\widehat{\mathbf{A}}_{\boldsymbol{\theta}}$ when the sample size is finite.

**Theorem 2.** *If $N \geq 2\epsilon^{-2} L_\beta^4 \log\left(2n^2/\delta\right)$, w.p. $1 - \beta - \delta$, we have $\frac{\left\|\widehat{\mathbf{A}}_{\boldsymbol{\theta}} - \mathbf{A}_{\boldsymbol{\theta}}\right\|_{\mathsf{F}}^2}{4\|\mathbf{A}_{\boldsymbol{\theta}}\|_{\mathsf{F}}^2} \leq \frac{\epsilon}{\mu}$.*

The proof of Theorem 2 is summarized in Appendix D. Theorem 2 guarantees that the estimation error $\left\|\widehat{\mathbf{A}}_{\boldsymbol{\theta}} - \mathbf{A}_{\boldsymbol{\theta}}\right\|_{\mathsf{F}}$ is bounded with high probability as a function of the desired accuracy $\epsilon$ and the number of samples $N$, provided that Assumption 2 holds. The bound contains the constant $4\|\mathbf{A}_{\boldsymbol{\theta}}\|_{\mathsf{F}}^2$, which depends on the true system matrix and is not directly accessible in practice. However, this

does not affect the generality or applicability of the result, since: (i) the constant $4\|\mathbf{A}_{\boldsymbol{\theta}}\|_{\mathsf{F}}^2$ is independent of the sample data and only scales the bound linearly, (ii) the rate of convergence is still determined by the sample size $N$ and the desired confidence level $\delta$, and (iii) in many practical scenarios, conservative upper bounds on $\|\mathbf{A}_{\boldsymbol{\theta}}\|_{\mathsf{F}}$ can be specified based on prior structural knowledge. We also note that the resulting sample complexity bound is conservative. This is primarily due to the worst-case nature of Assumption 2 and the use of union bounds in the probabilistic analysis. Nonetheless, the result provides a first-step theoretical understanding of the finite-sample behavior of our estimator and offers a guideline for selecting a sufficiently large snapshot dataset $\mathcal{D}_{\mathsf{snap}}$ to achieve a desired estimation accuracy.

### 4.3 RE-STABILIZATION

Optimal re-stabilization design considers the following problem:

$$
\begin{aligned}
\min_{\mathbf{B}_{\boldsymbol{\theta}},\, \mathbf{K}_{\boldsymbol{\theta}}} \quad & J_{\boldsymbol{\theta}}(\mathbf{x}(0)) := \int_0^{\infty} \mathbf{x}^{\top}(t)\mathbf{K}_{\boldsymbol{\theta}}^{\top}\mathbf{K}_{\boldsymbol{\theta}}\mathbf{x}(t)\mathrm{d}t,\ \forall \mathbf{x}(0) \\
\text{s.t.} \quad & \mathsf{Re}(\lambda_{\mathsf{d},\boldsymbol{\theta}}) - \mathsf{Re}(\tilde{\lambda}_{\mathsf{d},\boldsymbol{\theta}}) = \lambda_{\mathsf{s}} > 0,\ \mathbf{B}_{\boldsymbol{\theta}} \in \mathcal{B},\ \mathbf{K}_{\boldsymbol{\theta}} \in \mathcal{K}(\mathbf{B}_{\boldsymbol{\theta}}, \lambda_{\mathsf{s}}).
\end{aligned}
\tag{8}
$$

Here, $\mathcal{B}$ denotes the feasible set for the input assignment matrix $\mathbf{B}_{\boldsymbol{\theta}}$, and $\mathcal{K}(\mathbf{B}_{\boldsymbol{\theta}}, \lambda_{\mathsf{s}})$ denotes the admissible feedback gain matrices that ensure the desired re-stabilization margin $\lambda_{\mathsf{s}}$ is achieved. That is, the dominant eigenvalue of the original system, $\lambda_{\mathsf{d},\boldsymbol{\theta}}$, is shifted by $\lambda_{\mathsf{s}}$ in real part under the closed-loop dynamics. The input assignment $\mathbf{B}_{\boldsymbol{\theta}}$ and feedback gain $\mathbf{K}_{\boldsymbol{\theta}}$ obtained by solving equation 8 minimize the total input energy while enforcing the desired stabilization requirement for any initial state $\mathbf{x}(0)$. By Theorem 1 of (Yasukata et al., 2023), optimal solution is computed as $\mathbf{B}_{\boldsymbol{\theta}}^{\star} = \arg\max_{\mathbf{B} \in \mathcal{B}} \|\mathbf{w}_{\mathsf{d},\boldsymbol{\theta}}^{\top}\mathbf{B}\|$, $\mathbf{K}_{\boldsymbol{\theta}}^{\star} = -\lambda_{\mathsf{s}}\mathbf{B}_{\boldsymbol{\theta}}^{\star\top}\mathbf{v}_{\mathsf{d},\boldsymbol{\theta}}\mathbf{w}_{\mathsf{d},\boldsymbol{\theta}}^{\top}$. Here, $\mathbf{v}_{\mathsf{d},\boldsymbol{\theta}}$ and $\mathbf{w}_{\mathsf{d},\boldsymbol{\theta}}$ are the right and left eigenvectors corresponding to the dominant eigenvalue $\lambda_{\mathsf{d},\boldsymbol{\theta}}$ of the open-loop system. This formulation provides a computationally efficient solution while directly linking the input design to the spectral structure of the system. In particular, the dominant eigenvalue $\lambda_{\mathsf{d},\boldsymbol{\theta}}$ governs the direction in which stabilization is most critical, and the optimal assignment aligns control along the corresponding eigenvectors to ensure energy-efficient re-stabilization.

In practice, it is not possible to control the interactions of many proteins and mRNAs simultaneously. Thus, both assignment and feedback gain should be designed to be sparse. The selection is constrained such that $\mathbf{b}_{i,\boldsymbol{\theta}} \in \mathcal{E}_n$ for all $i = 1, \ldots, l$, where each $\mathbf{b}_{i,\boldsymbol{\theta}}$ must be distinct if $i \neq j$. The set $\mathcal{E}_n := \{\mathbf{e}_1, \ldots, \mathbf{e}_n\}$ corresponds to interventions applied to mRNAs, while $\mathcal{E}_n := \{\mathbf{e}_{n+1}, \ldots, \mathbf{e}_{2n}\}$ corresponds to protein-level interventions. Here, $\mathbf{e}_i$ denotes the $i$-th standard basis vector in $\mathbb{R}^{2n}$. In each intervention strategy, we assume either mRNA or protein-level intervention is selected, but not both simultaneously. The feasible set of input placements is denoted by $\mathcal{B}_l$. The gain matrix $\mathbf{K}_{\boldsymbol{\theta}} = [\mathbf{k}_{1,\boldsymbol{\theta}}, \ldots, \mathbf{k}_{l,\boldsymbol{\theta}}]^{\top}$ determines how each selected gene is perturbed. We constrain $\mathbf{k}_i(\boldsymbol{\theta}) = \kappa_i \mathbf{b}_{i,\boldsymbol{\theta}}$ so that the feedback acts only along the direction of the selected intervention site. The corresponding feasible set is denoted by $\mathcal{K}_l$. Under these constraints, the feedback only alters the diagonal elements of the system matrix $\mathbf{A}_{\boldsymbol{\theta}}$. This reflects a realistic intervention model in gene regulation: for instance, RNA interference and gene overexpression typically modulate only the self-dynamics (self-loops) of individual genes (Meister et al., 2013), corresponding to diagonal entries. Diagonal re-stabilization is defined as follows.

**Definition 1.** *Given an intervention budget $l < n$, diagonal re-stabilization refers to the design of $\mathbf{B}_{\boldsymbol{\theta}} \in \mathcal{B}_l$ and $\mathbf{K}_{\boldsymbol{\theta}} \in \mathcal{K}_l$ such that the dominant eigenvalue of the closed-loop system $\tilde{\lambda}_{\mathsf{d},\boldsymbol{\theta}} := \max \mathsf{Re}\left(\mathsf{eig}\left(\mathbf{A}_{\boldsymbol{\theta}} + \mathbf{B}_{\boldsymbol{\theta}}\mathbf{K}_{\boldsymbol{\theta}}\right)\right)$ satisfies $\mathsf{Re}(\tilde{\lambda}_{\mathsf{d},\boldsymbol{\theta}}) < \mathsf{Re}(\lambda_{\mathsf{d},\boldsymbol{\theta}})$.*

For a given input placement $\mathbf{B}_{\boldsymbol{\theta}} \in \mathcal{B}_l$, let $I_{\mathbf{B}_{\boldsymbol{\theta}}}$ be an input-index set defined as: $I_{\mathbf{B}_{\boldsymbol{\theta}}} := \{i \in \{1, \ldots, 2n\} : \exists j \in \{1, \ldots, l\}, \mathbf{b}_{j,\boldsymbol{\theta}} = \mathbf{e}_i\}$. Let $I_{\mathsf{vec},\mathbf{B}_{\boldsymbol{\theta}}}$ be a vector formed by extracting the elements from $I_{\mathbf{B}_{\boldsymbol{\theta}}}$: $I_{\mathsf{vec},\mathbf{B}_{\boldsymbol{\theta}}} = [s_1, \ldots, s_l]^{\top}$, where $s_i \in I_{\mathbf{B}_{\boldsymbol{\theta}}}$, $i = 1, \ldots, l$, and assume, without loss of generality, that $s_1 < s_2 < \ldots < s_l$. Diagonal re-stabilization for large-scale network systems has been addressed in (Shen et al., doi:10.1109/TETCI.2024.3442824). We summarize (Shen et al., doi:10.1109/TETCI.2024.3442824, Theorems 3 and 4) for our setting as the following lemma.

**Lemma 1.** *The system matrix becomes $\mathbf{A}_{\boldsymbol{\theta}} + \mathbf{B}_{\boldsymbol{\theta}}\mathbf{K}_{\boldsymbol{\theta}}$ after incorporating $\mathbf{B}_{\boldsymbol{\theta}} \in \mathcal{B}_l$ and $\mathbf{K}_{\boldsymbol{\theta}} \in \mathcal{K}_l$. The first-order approximation with respect to the Frobenius norm $\|\mathbf{B}_{\boldsymbol{\theta}}\mathbf{K}_{\boldsymbol{\theta}}\|_2$ of the dominant eigenvalue*

$\tilde{\lambda}_{\mathsf{d},\boldsymbol{\theta}}$ *of the matrix* $\mathbf{A}_{\boldsymbol{\theta}} + \mathbf{B}_{\boldsymbol{\theta}}\mathbf{K}_{\boldsymbol{\theta}}$ *is given by* $\tilde{\lambda}_{\mathsf{d},\boldsymbol{\theta}}^{(1)} = \lambda_{\mathsf{d},\boldsymbol{\theta}} + \frac{\sum_{i=1}^{l} \kappa_i \mathbf{w}_{\mathsf{d},\boldsymbol{\theta}}(s_i)\mathbf{v}_{\mathsf{d},\boldsymbol{\theta}}(s_i)}{\mathbf{w}_{\mathsf{d},\boldsymbol{\theta}}^{\top}\mathbf{v}_{\mathsf{d},\boldsymbol{\theta}}}$. *If the input placement* $\mathbf{B}_{\boldsymbol{\theta}}$ *includes a key gene, then diagonal re-stabilization can be achieved.*

From Lemma 1, we see that diagonal re-stabilization serves as a tractable approximation of the optimal re-stabilization design. In particular, selecting input locations that maximize the absolute value of the product $\mathbf{w}_{\mathsf{d},\boldsymbol{\theta}}(s_i)\mathbf{v}_{\mathsf{d},\boldsymbol{\theta}}(s_i)$ enhances control effectiveness, since these terms directly influence the leading-order eigenvalue shift.

**Theorem 3.** *Assume that the maximum eigenvalue* $\lambda_{\mathsf{d}}$ *of* $\mathbf{A}_{\boldsymbol{\theta}}$ *is with eigengap* $\delta_{\lambda} = \min_{i=2,\dots,n} |\lambda_{\mathsf{d}} - \lambda_i| > 0$. *If* $N \geq 2\epsilon^{-2} L_{\beta}^4 \log\left(2n^2/\delta\right)$, *with probability* $1 - \beta - \delta$, *we have* $\|\widehat{\mathbf{w}}_{\mathsf{d}} - \mathbf{w}_{\mathsf{d}}\|_2 \leq \frac{L}{\mu\,\delta_{\lambda}}\epsilon$.

Proof of Theorem 3 is summarized in Appendix E. Theorem 3 establishes a finite-sample probabilistic bound on the error between the estimated dominant left eigenvector $\widehat{\mathbf{w}}_{\mathsf{d}}$ and the true dominant eigenvector $\mathbf{w}_{\mathsf{d}}$ of the system matrix $\mathbf{A}_{\boldsymbol{\theta}}$. The bound holds with high probability $1 - \beta - \delta$, provided that the sample size $N$ is sufficiently large. Notably, the error bound scales inversely with both the strong convexity modulus $\mu$ and the eigengap $\delta_{\lambda}$, highlighting that the estimation becomes more reliable when the spectrum of $\mathbf{A}_{\boldsymbol{\theta}}$ exhibits clear separation between the dominant eigenvalue and the others. This result plays a critical role in ensuring the accuracy of approximate diagonal re-stabilization design. As shown in Lemma 1, the first-order approximation of the stabilized dominant eigenvalue $\tilde{\lambda}_{\mathsf{d},\boldsymbol{\theta}}^{(1)}$ depends on the product $\mathbf{w}_{\mathsf{d},\boldsymbol{\theta}}(s_i)\mathbf{v}_{\mathsf{d},\boldsymbol{\theta}}(s_i)$ at selected indices $s_i$. Therefore, accurate estimation of $\mathbf{w}_{\mathsf{d},\boldsymbol{\theta}}$ directly affects the effectiveness of the control input selection strategy. The finite-sample guarantee in Theorem 3 ensures that the estimated eigenvector $\widehat{\mathbf{w}}_{\mathsf{d}}$ remains sufficiently close to the true one, thus preserving the control relevance of the index selection even under sampling uncertainty. In practical terms, this means that even when the snapshot data set is limited in size, the approximate diagonal re-stabilization strategy can still reliably identify high-impact nodes—e.g., key genes or transcription factor, by leveraging the structure encoded in $\widehat{\mathbf{w}}_{\mathsf{d}}$. Although the theoretical bound may be conservative due to its dependence on worst-case assumptions, it provides a principled basis for quantifying confidence in structure-aware control from finite data.

## 5 VALIDATIONS

**Simulation model.** To validate our framework for system identification and early treatment design, we conduct numerical experiments on synthetic mRNA-protein regulatory networks that mimic disease progression through bifurcation. We focus on a global bifurcation scenario, where a master regulator protein controls the transcription rates of all genes, leading to a system-wide tipping point. This setup aligns with biological systems exhibiting coordinated dysregulation, such as in cancer or developmental disorders, where master regulators drive cell fate transitions via bistable switches. For noise, we employ additive Gaussian perturbations to reflect extrinsic biological variability, consistent with the assumptions in our theoretical analysis (e.g., covariance estimation in Section 2.3). The model simulates a system of 5 genes ($G_0, \dots, G_4$) and their corresponding 5 proteins ($P_0, \dots, P_4$). The network topology is designed around a single master regulator, protein $P_4$, which positively regulates the expression of all genes in the network, including its own. This global positive feedback structure allows the system to switch between low and high expression states, characteristic of bistability. The dynamics of the network are modeled as a system of stochastic differential equations (SDEs) to capture the intrinsic noise inherent in biological processes. We specifically use an additive Gaussian noise model, where the stochastic fluctuations are independent of the molecular concentrations. The transition between system states is induced by varying a key bifurcation parameter, $K_{\text{master}}$, which controls the activation threshold of the master regulator. This setup allows us to generate high-dimensional, single-cell snapshot data at various points along the system's trajectory as it approaches the tipping point. For a comprehensive description of the network equations, noise setting, and a full list of parameter values, please check Appendix F.

**Results.** According to Lemma 1, the value of $\mathbf{w}_{\mathsf{d},\boldsymbol{\theta}}(i)\mathbf{v}_{\mathsf{d},\boldsymbol{\theta}}(i)$ plays a critical role in the diagonal re-stabilization design. The most effective node is the one corresponding to the maximum absolute value of this product. Moreover, the sign of $\mathbf{w}_{\mathsf{d},\boldsymbol{\theta}}(s_i)\mathbf{v}_{\mathsf{d},\boldsymbol{\theta}}(s_i)$ is also essential. If the sign is positive, a negative $k_i$ is required to shift the real part of the dominant eigenvalue further from zero. Conversely, if the sign is negative, a positive $k_i$ is needed. Owing to these considerations, our analysis

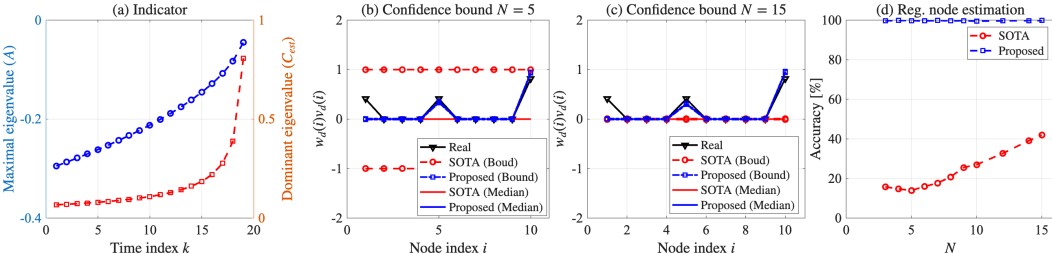

Figure 3: Plots of the evolution of eigenvalues across episodes and the estimation results: (a) Maximum eigenvalues of $\mathbf{A}_{\boldsymbol{\theta}}$ and dominant eigenvalues of $\mathbf{C}_{\boldsymbol{\theta}}$ at different time points; (b) 95% confidence region of the estimated $\mathbf{w}_{\mathrm{d},\boldsymbol{\theta}_{19}}(i)\mathbf{v}_{\mathrm{d},\boldsymbol{\theta}_{19}}(i)$ with $N = 5$; (c) 95% confidence region of the estimated $\mathbf{w}_{\mathrm{d},\boldsymbol{\theta}_{19}}(i)\mathbf{v}_{\mathrm{d},\boldsymbol{\theta}_{19}}(i)$ with $N = 15$. (d) Statistical analysis of estimation accuracy for the regulatory node with results of 2,000 trials.

focuses on the estimation accuracy of $\mathbf{w}_{\mathrm{d},\boldsymbol{\theta}}(s_i)\mathbf{v}_{\mathrm{d},\boldsymbol{\theta}}(s_i)$ in the subsequent discussion. In particular, we evaluate the percentage of trials in which the method correctly identifies the regulatory node with the largest absolute value of $\mathbf{w}_{\mathrm{d},\boldsymbol{\theta}}(s_i)\mathbf{v}_{\mathrm{d},\boldsymbol{\theta}}(s_i)$, along with the correct sign. This metric directly reflects the method's ability to support effective diagonal re-stabilization, as discussed in Lemma 1. We perform a Monte Carlo simulation to investigate the statistical performance of the proposed method. In each simulation, the network's structure and parameters have been fixed. The cases with sample number as $N = 3, 5, 7, 8, 9, 10, 12, 15$ were considered. We have totally $H = 19$ episodes for snapshot data observations. On the other hand, for each sample number $N$, 2000 sample sets were generated. We implement the method in (Shen et al., 2024) (SOTA) and the proposed method (Proposed) to obtain the system matrix estimations. As the episode proceeds from $k = 1$ to $k = 19$, the maximum eigenvalue of the system matrix gradually approaches zero, and the covariance matrix increases rapidly, as shown in Figure 3 (a). Figures 3 (b) and (c) show the 95% confidence bounds of the estimated $\mathbf{w}_{\mathrm{d},\boldsymbol{\theta}_{19}}(i)\mathbf{v}_{\mathrm{d},\boldsymbol{\theta}_{19}}(i)$ obtained by the SOTA method and the proposed method with $N = 5 < 2n = 10$ and $N = 15 > 2n = 10$, respectively. The proposed method provides confidence bounds that are close to the true value, while the SOTA method fails to yield accurate estimations. The results of 2,000 Monte Carlo trials on the regulatory node estimations are summarized in Figure 3 (d). As the number of samples increases, both algorithms exhibit improved accuracy of identifying the regulatory node along with the correct sign of . Notably, the proposed method demonstrates significant improvements in regulatory node estimation accuracy compared to the state-of-the-art (SOTA) approach. In particular, when the number of samples is considerably smaller than the system state dimension $2n = 10$, the proposed method still achieves high estimation accuracy. The above performance gain arises from the method's ability to leverage information across multiple episodes, thereby enriching the data available for covariance matrix estimation. Consequently, this enhancement leads to a more accurate estimation of the system matrix and its associated eigenvectors. In this simulation, we focus on the identification of regulatory genes and do not present intervention results. This is because, once the key regulatory nodes are accurately identified, the corresponding intervention strategies—such as re-stabilization via feedback—can be effectively designed based on existing control-theoretic formulations. Therefore, the success of the overall intervention critically depends on the accuracy of the regulatory gene identification, which is the main focus of this work.

## 6    CONCLUSIONS

We proposed a system identification framework for mRNA-protein regulatory networks from snapshot data, tailored to design effective intervention strategies. By exploiting the Lyapunov equation with structural constraints, our method achieves reliable estimation of the system matrix under high-dimensional low-sample-size conditions, with theoretical guarantees on finite-sample accuracy. Building on these results, we developed both optimal and approximate diagonal re-stabilization designs, offering actionable insights for early treatment at the pre-disease stage. Future work will extend this framework to nonlinear dynamics and validate intervention strategies on real single-cell datasets, with the potential to impact broader applications in biology and beyond.

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

## Appendix

## A  EARLY DETECTION OF PRE-DISEASE STAGE

Let $\mathbf{C}_{\boldsymbol{\theta}} \in \mathbb{R}^{2n \times 2n}$ denote the covariance matrix of $\mathbf{x}$ when the system is parameterized by $\boldsymbol{\theta}$. As presented in (Oku & Aihara, 2018, Section 3.2), as the parameter $\boldsymbol{\theta}$ approaches the tipping point $\boldsymbol{\theta}_{\mathsf{C}}$, the covariance matrix $\mathbf{C}_{\boldsymbol{\theta}}$ converges to a rank-one matrix if $\lambda_{\mathsf{d},\boldsymbol{\theta}}$ is real, written by

$$\lim_{\lambda_{\mathsf{d},\boldsymbol{\theta}} \to 0^-} (-2\lambda_{\mathsf{d},\boldsymbol{\theta}}) \mathbf{C}_{\boldsymbol{\theta}} = \widetilde{D}_{11} \mathbf{v}_{\mathsf{d},\boldsymbol{\theta}} \mathbf{v}_{\mathsf{d},\boldsymbol{\theta}}^{\top}, \tag{9}$$

where the positive real number $\widetilde{D}_{11}$ is the first diagonal element of $\widetilde{\mathbf{D}} = \mathbf{V}_{\boldsymbol{\theta}}^{-1} \mathbf{D} (\mathbf{V}_{\boldsymbol{\theta}}^{\mathsf{H}})^{-1}$. Here, $\mathbf{V}_{\boldsymbol{\theta}}^{\mathsf{H}}$ is a matrix consisting all right eigenvectors of $\mathbf{C}_{\boldsymbol{\theta}}$. On the other hand, the covariance matrix $\mathbf{C}_{\boldsymbol{\theta}}$ converges to a rank-two matrix if $\lambda_{\mathsf{d},\boldsymbol{\theta}}$ is complex. Specifically, the dominant eigenvalue of $\mathbf{C}_{\boldsymbol{\theta}}$'s limit is infinity.

A group of diagonal elements of $\mathbf{C}_{\boldsymbol{\theta}}$ also increases to infinity, indicating that there exists a group of nodes whose standard deviations become unbounded. In parallel, the covariance between two nodes in this group also grows substantially as $\boldsymbol{\theta}$ approaches the tipping point $\boldsymbol{\theta}_{\mathsf{C}}$. We take the case when $\lambda_{\mathsf{d},\boldsymbol{\theta}}$ as an example to illustrate it. The eigenvector $\boldsymbol{\xi}_{\mathsf{d},\boldsymbol{\theta}} = [\boldsymbol{\xi}_{\mathsf{d},\boldsymbol{\theta},\mathsf{r}}^{\top}, \boldsymbol{\xi}_{\mathsf{d},\boldsymbol{\theta},\mathsf{p}}^{\top}]^{\top}$ of $\mathbf{C}_{\boldsymbol{\theta}}$ corresponding to the dominant eigenvalue converges to the right MAC eigenvector of $\mathbf{A}_{\boldsymbol{\theta}}$, namely,

$$\lim_{\lambda_{\mathsf{d},\boldsymbol{\theta}} \to 0^-} \boldsymbol{\xi}_{\mathsf{d},\boldsymbol{\theta}} = \mathbf{v}_{\mathsf{d},\boldsymbol{\theta}}. \tag{10}$$

By equation 9, if $\mathbf{v}_{\mathsf{d},\boldsymbol{\theta}}(i) \neq 0$ and $\mathbf{v}_{\mathsf{d},\boldsymbol{\theta}}(j) \neq 0$, we have

$$\lim_{\lambda_{\mathsf{d},\boldsymbol{\theta}} \to 0^-} |\mathbf{C}_{\boldsymbol{\theta}}(i,j)| = \infty, \tag{11}$$

which is the theoretical explanation for the system's large fluctuations in the pre-disease stage. The standard deviation corresponds to the diagonal element $\mathbf{C}_{\boldsymbol{\theta}}(i,i) \approx \mathbf{v}_{\mathsf{d},\boldsymbol{\theta}}^2(i)$. Consequently, the standard deviation of the node, where $\mathbf{v}_{\mathsf{d},\boldsymbol{\theta}}(i) \neq 0$, shows a significant increase at the pre-disease stage. The covariance between two nodes ($\mathbf{v}_{\mathsf{d},\boldsymbol{\theta}}(i) \neq 0$, $\mathbf{v}_{\mathsf{d},\boldsymbol{\theta}}(j) \neq 0$) also increases significantly in the pre-disease stage.

The above two properties provide a theoretical explanation for the system's large fluctuations in the pre-disease stage, when $\boldsymbol{\theta} \approx \boldsymbol{\theta}_{\mathsf{C}}$. By examining system fluctuations from snapshot data, it becomes possible to detect the pre-disease stage, which is completely model-free. We refer to (Chen et al., 2012; Aihara et al., 2022) for more details about indicator selection and computation for pre-disease detection by snapshot data.

## B  RANDOM VARIABLE IN AN AUGMENTED SPACE.

Consider a random variable $\boldsymbol{\xi} := (\mathbf{X}, T)$ with support $\Xi \subset \mathbb{R}^{n+1}$, where $\xi := (\mathbf{x}, t)^2$ is a realization of $\boldsymbol{\xi}$. Let $\mathcal{F}$ denote the $\sigma$-algebra of subsets of $\Xi$. Equipped with a probability measure $\rho$ defined on the Borel space $(\Xi, \mathcal{F})$, this forms a probability space $(\Xi, \mathcal{F}, \rho)$. Let $\Xi_{\mathsf{s}} \subseteq \Xi$ be a subset of $\Xi$. Given a continuous probability density function $p(\xi)$ with support $\Xi$, the probability that the random variable $\boldsymbol{\xi}$ lies within $\Xi_{\mathsf{s}}$ is expressed as $\mathsf{Pr}\{\boldsymbol{\xi} \in \Xi_{\mathsf{s}}\} := \int_{\Xi_{\mathsf{s}}} p(\xi) \, \mathrm{d}\xi$. Suppose that the probability density function $p(\xi)$ is a joint density denoted by $p(\mathbf{x}, t)$. Both $\mathbf{x}$ and $t$ are continuous random variables with marginal probability densities[3] given by $p_{\mathbf{X}}(\mathbf{x}) := \int_{\mathbb{R}^+} p(\mathbf{x}, t) \, \mathrm{d}t$, $p_T(t) := \int_{\mathbb{R}^n} p(\mathbf{x}, t) \, \mathrm{d}\mathbf{x}$. Consider that $p_T(t) > 0$ and $p(\mathbf{x}, t) > 0$ hold for any $\mathbf{x} \in \mathbf{X}$ and any $(\mathbf{x}, t) \in \mathbb{R}^{n+1}$. A conditional probability density is defined by $p_{\mathbf{X}}^{\mathsf{c}}(\mathbf{x}|T = t) := p(\mathbf{x}, t) / p_T(t)$. We omit "$T =$" in the conditional probability density in the remainder of the paper. Additionally, we also define the conditional probability of having $\mathbf{X} \in \mathcal{X}_{\mathsf{s}} \subseteq \mathbb{R}^n$ when $T = t$ by $\mathsf{Pr}\{\mathbf{X} \in \mathcal{X}_{\mathsf{s}}|t\} := \int_{\mathcal{X}_{\mathsf{s}}} p_{\mathbf{X}}^{\mathsf{c}}(\mathbf{x}|t) \mathrm{d}\mathbf{x}$.

---

[2] $\xi_k := \left( \mathbf{x}_m^{(k)}, t_k \right)$ can represent a data point from the snapshot dataset.

[3] Although the time point or sample time index may appear as a discrete integer, it is treated in a general sense as a continuous variable. Moreover, a continuous probability density is used to approximate discrete probability over a continuous domain (Capinski & Kopp, 2004).

## C    PROOF OF THEOREM 1

As a preparation, we first introduce the convergence of

**Theorem 4.** *Then, as $N, H \to \infty$, we have $\widehat{\mathbf{C}}_{\boldsymbol{\theta}} \xrightarrow{p} \mathbf{C}_{\boldsymbol{\theta}}$.*

*Proof.* The proof consists of two steps.

**Step 1: Convergence of $\widehat{p}^{\mathsf{c}}_{\mathbf{X}}(\mathbf{x} \mid t_k)$ to $p^{\mathsf{c}}_{\mathbf{X}}(\mathbf{x} \mid t_k)$, and convergence of corresponding covariance.**

By the results in (Gooijer & Zerom, 2003; Hall et al., 1999), under the stated bandwidth conditions $h \to 0$ and $NHh^n \to \infty$, the NW conditional density estimator satisfies

$$\sup_{\mathbf{x} \in \mathbb{R}^n} |\widehat{p}^{\mathsf{c}}_{\mathbf{X}}(\mathbf{x} \mid t_k) - p^{\mathsf{c}}_{\mathbf{X}}(\mathbf{x} \mid t_k)| \xrightarrow{p} 0 \quad \text{as} \quad N, H \to \infty.$$

Since the true density $p^{\mathsf{c}}_{\mathbf{X}}$ is Gaussian $\mathcal{N}(\mathbf{0}^n, \mathbf{C}_{\boldsymbol{\theta}})$ with finite second moments, and the uniform convergence holds, we can invoke standard results on the convergence of integrals of bounded functions with respect to densities (e.g., Theorem 2.1 in (Gooijer & Zerom, 2003)).

Let $g(\mathbf{x}) = \mathbf{x}\mathbf{x}^{\top}$, which is polynomially bounded. Then, as $\quad N, H \to \infty$,

$$\int g(\mathbf{x})\widehat{p}^{\mathsf{c}}_{\mathbf{X}}(\mathbf{x} \mid t_k)d\mathbf{x} \xrightarrow{p} \int g(\mathbf{x})p^{\mathsf{c}}_{\mathbf{X}}(\mathbf{x} \mid t_k)d\mathbf{x} = \mathbf{C}_{\boldsymbol{\theta}}. \tag{12}$$

Thus, the covariance matrix of $\widehat{p}^{\mathsf{c}}_{\mathbf{X}}$ converges in probability to $\mathbf{C}_{\boldsymbol{\theta}}$.

**Step 2: Convergence of empirical covariance $\widehat{\mathbf{C}}_{\boldsymbol{\theta}}$ to the covariance of $\widehat{p}^{\mathsf{c}}_{\mathbf{X}}$.**

Now consider $\{\widehat{\mathbf{x}}^{(k)}_m\}^M_{m=1}$ as i.i.d. samples from $\widehat{p}^{\mathsf{c}}_{\mathbf{X}}(\mathbf{x} \mid t_k)$. By the Weak Law of Large Numbers, for any $\epsilon_1 > 0$ and $\delta_1 > 0$, there exists $M_0$ such that for all $M \geq M_0$, we have

$$\Pr\left\{\left\|\widehat{\mathbf{C}}_{\boldsymbol{\theta}} - \int \mathbf{x}\mathbf{x}^{\top}\widehat{p}^{\mathsf{c}}_{\mathbf{X}}(\mathbf{x} \mid t_k)d\mathbf{x}\right\|_{\mathsf{F}} < \epsilon_1\right\} \geq 1 - \delta_1. \tag{13}$$

**Combination of the two steps: $\epsilon$-$\delta$ argument.**

Finally, we combine the two steps carefully. Let $\epsilon > 0$ and $\delta > 0$ be arbitrary.

- From Step 1, there exists $(N_0, H_0)$ such that for all $N \geq N_0, H \geq H_0$,

$$\Pr\left\{\left\|\int \mathbf{x}\mathbf{x}^{\top}\widehat{p}^{\mathsf{c}}_{\mathbf{X}}(\mathbf{x} \mid t_k)d\mathbf{x} - \mathbf{C}_{\boldsymbol{\theta}}\right\|_{\mathsf{F}} < \epsilon/2\right\} \geq 1 - \delta/2.$$

- From Step 2, for these fixed $(N, H)$ and for sufficiently large $M \geq M_0$, we have

$$\Pr\left\{\left\|\widehat{\mathbf{C}}_{\boldsymbol{\theta}} - \int \mathbf{x}\mathbf{x}^{\top}\widehat{p}^{\mathsf{c}}_{\mathbf{X}}(\mathbf{x} \mid t_k)d\mathbf{x}\right\|_{\mathsf{F}} < \epsilon/2\right\} \geq 1 - \delta/2.$$

Now applying the union bound, for $N \geq N_0, H \geq H_0, M \geq M_0$, we obtain:

$$\Pr\left\{\left\|\widehat{\mathbf{C}}_{\boldsymbol{\theta}} - \mathbf{C}_{\boldsymbol{\theta}}\right\|_{\mathsf{F}} < \epsilon\right\} \geq 1 - \delta.$$

Since $\epsilon$ and $\delta$ are arbitrary, we conclude that:

$$\widehat{\mathbf{C}}_{\boldsymbol{\theta}} \xrightarrow{p} \mathbf{C}_{\boldsymbol{\theta}} \quad \text{as} \quad M, N, H \to \infty.$$

$\square$

By Theorem 4, for any given $\mathbf{A}_{\boldsymbol{\theta}}$, it holds that

$$J\left(\mathscr{A}, \widehat{\mathbf{C}}_{\boldsymbol{\theta}}\right) \xrightarrow{p} J\left(\mathscr{A}, \mathbf{C}_{\boldsymbol{\theta}}\right), \quad \text{as } N, H \to \infty. \tag{14}$$

That is, for any fixed $\mathscr{A}$, the objective function evaluated with the estimated covariance converges in probability to its true counterpart. By invoking (Shapiro et al., 2014, Proposition 5.1), this pointwise convergence on a compact set implies that

$$J\left(\mathscr{A}, \widehat{\mathbf{C}}_{\boldsymbol{\theta}}\right) \to J\left(\mathscr{A}, \mathbf{C}_{\boldsymbol{\theta}}\right) \tag{15}$$

holds uniformly on any compact subset of $\mathbb{R}^{3n+n^2}$, where $\mathbb{R}^{3n+n^2}$ denotes the parameter space of all vectors containing the nonzero elements of $\mathbf{A}$. Then, by (Shapiro et al., 2014, Proposition 5.2), the convergence of the objective functions guarantees that the minimizer of equation $\widehat{\mathcal{P}}_{\boldsymbol{\theta}}\left(\mathcal{D}_{\text{snap}}\right)$ converges in probability to the minimizer of equation $\mathcal{P}_{\boldsymbol{\theta}}$ as $N, H \to \infty$. Since equation $\mathcal{P}_{\boldsymbol{\theta}}$ admits a unique optimal solution $\mathbf{A}_{\boldsymbol{\theta}}$, it follows that

$$\widehat{\mathbf{A}}_{\boldsymbol{\theta}} \xrightarrow{p} \mathbf{A}_{\boldsymbol{\theta}} \tag{16}$$

as $N, H \to \infty$, which completes the proof.

# D    PROOF OF THEOREM 2

As preparation of the proof, we first give the following results regarding the confidence bound on the covariance matrix estimation.

**Theorem 5.** *Suppose Assumption 2 holds for some $L_\beta$ and $\beta \in (0,1)$. Let $\alpha \in (0,1)$ and $\epsilon > 0$ be given.*

*Then, for any $\delta \in (0,1)$, if the number of generated samples $M$ satisfies:*

$$M \geq \frac{2L_\beta^4 \log\left(\frac{2n^2}{\delta}\right)}{\epsilon^2},$$

*then with probability at least $1 - \beta - \delta$, we have:*

$$\left\|\widehat{\mathbf{C}}_{\boldsymbol{\theta}} - \mathbf{C}_{\boldsymbol{\theta}}\right\|_{\mathsf{F}} \leq \epsilon.$$

*Proof.* Let $\epsilon > 0$ and $\delta \in (0,1)$ be given.

We split the error $\left\|\widehat{\mathbf{C}}_{\boldsymbol{\theta}} - \mathbf{C}_{\boldsymbol{\theta}}\right\|_{\mathsf{F}}$ into two terms:

$$\left\|\widehat{\mathbf{C}}_{\boldsymbol{\theta}} - \mathbf{C}_{\boldsymbol{\theta}}\right\|_{\mathsf{F}} \leq \underbrace{\left\|\widehat{\mathbf{C}}_{\boldsymbol{\theta}} - \mathbb{E}_{\widehat{p}}[\mathbf{x}\mathbf{x}^\top]\right\|_{\mathsf{F}}}_{X_1} + \underbrace{\left\|\mathbb{E}_{\widehat{p}}[\mathbf{x}\mathbf{x}^\top] - \mathbf{C}_{\boldsymbol{\theta}}\right\|_{\mathsf{F}}}_{X_2}.$$

**Step 1: Control of $X_2$.**

From Theorem 4 Step 1, we know that as $N, H \to \infty$:

$$\left\|\mathbb{E}_{\widehat{p}}[\mathbf{x}\mathbf{x}^\top] - \mathbf{C}_{\boldsymbol{\theta}}\right\|_{\mathsf{F}} \xrightarrow{p} 0.$$

Therefore, there exists $(N_0, H_0)$ such that for $N \geq N_0, H \geq H_0$, we have:

$$\Pr\left\{X_2 \leq \epsilon/2\right\} \geq 1 - \delta/2.$$

**Step 2: Control of $X_1$ using Hoeffding inequality.**

Under Assumption 2, for each entry $(i, j)$ of the matrix $\mathbf{x}\mathbf{x}^\top$, we have:

$$|x^{(i)}x^{(j)}| \leq L_\beta^2 \quad \text{with probability at least } 1 - \beta.$$

Conditioning on the event where this bound holds, we apply Hoeffding inequality (Hoeffding, 1963) to each entry of the matrix. For each $(i, j)$:

$$\Pr\left\{\left|\frac{\sum_{m=1}^{M} \mathbf{x}_m^{(i)} \mathbf{x}_m^{(j)} - \mathbb{E}_{\widehat{p}}^{i,j}}{M}\right| \geq \epsilon'\right\} \leq 2 \exp\left(-\frac{2M\epsilon'^2}{4L_\beta^4}\right), \tag{17}$$

where

$$\mathbb{E}_{\widehat{p}}^{i,j} := \mathbb{E}_{\widehat{p}}[\mathbf{x}^{(i)} \mathbf{x}^{(j)}]. \tag{18}$$

Now set:

$$\epsilon' := \frac{\epsilon}{2n}.$$

Then applying union bound over $n^2$ matrix entries:

$$\Pr\left\{X_1 \leq \epsilon/2\right\} \geq 1 - 2n^2 \exp\left(-\frac{2M\epsilon^2}{4n^2 L_\beta^4}\right).$$

To ensure this probability is at least $1 - \delta/2$, it suffices to choose $M$ such that:

$$2n^2 \exp\left(-\frac{M\epsilon^2}{2n^2 L_\beta^4}\right) \leq \delta/2.$$

Solving this inequality yields the stated bound:

$$M \geq \frac{2L_\beta^4 \log\left(\frac{2n^2}{\delta}\right)}{\epsilon^2}.$$

**Final step: Union bound.**

Finally, applying union bound over the two steps, we obtain:

$$\Pr\left\{\left\|\widehat{\mathbf{C}}_{\boldsymbol{\theta}} - \mathbf{C}_{\boldsymbol{\theta}}\right\|_{\mathsf{F}} \leq \epsilon\right\} \geq 1 - \beta - \delta.$$

$\square$

*Proof.* **Step 1: Sample covariance approximation.** By Theorem 5, for sufficiently large $M$, we have with probability at least $1 - \beta - \delta$,

$$\left\|\widehat{\mathbf{C}}_{\boldsymbol{\theta}} - \mathbf{C}_{\boldsymbol{\theta}}\right\|_{\mathsf{F}} \leq \epsilon.$$

**Step 2: Perturbation of optimal solution $\widehat{\mathbf{A}}_{\boldsymbol{\theta}}$.** Define the perturbed objective

$$J(\mathbf{A}, \mathbf{C}) := \left\|\mathbf{A}\mathbf{C} + \mathbf{C}\mathbf{A}^\top + \mathbf{D}\right\|_{\mathsf{F}}^2.$$

Since $\mathbf{A}_{\boldsymbol{\theta}}$ satisfies the Lyapunov equation for $\mathbf{C}_{\boldsymbol{\theta}}$, we have

$$\mathbf{Q}(\mathbf{A}_{\boldsymbol{\theta}}, \mathbf{C}_{\boldsymbol{\theta}}) = \mathbf{A}_{\boldsymbol{\theta}}\mathbf{C}_{\boldsymbol{\theta}} + \mathbf{C}_{\boldsymbol{\theta}}\mathbf{A}_{\boldsymbol{\theta}}^\top + \mathbf{D} = \mathbf{0}.$$

Thus, the first-order change of $J(\mathbf{A}_{\boldsymbol{\theta}}, \widehat{\mathbf{C}}_{\boldsymbol{\theta}})$ around $\mathbf{C}_{\boldsymbol{\theta}}$ is:

$$\mathbf{Q}(\mathbf{A}_{\boldsymbol{\theta}}, \widehat{\mathbf{C}}_{\boldsymbol{\theta}}) = \mathbf{A}_{\boldsymbol{\theta}}\Delta\mathbf{C} + \Delta\mathbf{C}\mathbf{A}_{\boldsymbol{\theta}}^\top,$$

$$\left\|\mathbf{Q}(\mathbf{A}_{\boldsymbol{\theta}}, \widehat{\mathbf{C}}_{\boldsymbol{\theta}})\right\|_F \leq 2\|\mathbf{A}_{\boldsymbol{\theta}}\|_F \|\Delta\mathbf{C}\|_F.$$

So the increase in objective function is bounded by:

$$J(\mathbf{A}_{\boldsymbol{\theta}}, \widehat{\mathbf{C}}_{\boldsymbol{\theta}}) = \left\|\mathbf{A}_{\boldsymbol{\theta}}\Delta\mathbf{C} + \Delta\mathbf{C}\mathbf{A}_{\boldsymbol{\theta}}^\top\right\|_F^2 \leq 4\|\mathbf{A}_{\boldsymbol{\theta}}\|_F^2 \|\Delta\mathbf{C}\|_F^2.$$

Letting
$$L := 4\|\mathbf{A_\theta}\|_F^2,$$
we obtain:
$$|J(\mathbf{A_\theta}, \widehat{\mathbf{C}}_\theta) - J(\mathbf{A_\theta}, \mathbf{C_\theta})| \le L\epsilon^2.$$

Now, from standard perturbation results for strongly convex objectives, the minimizer satisfies
$$\left\|\widehat{\mathbf{A}}_\theta - \mathbf{A_\theta}\right\|_F \le \frac{1}{\mu}\left\|\nabla_\mathbf{A} J(\mathbf{A_\theta}, \widehat{\mathbf{C}}_\theta)\right\|_F.$$

We further bound:
$$\nabla_\mathbf{A} J(\mathbf{A_\theta}, \widehat{\mathbf{C}}_\theta) = 2\left(\mathbf{A_\theta}\widehat{\mathbf{C}}_\theta\widehat{\mathbf{C}}_\theta + \widehat{\mathbf{C}}_\theta\mathbf{A_\theta}^\top\widehat{\mathbf{C}}_\theta + \mathbf{D}\widehat{\mathbf{C}}_\theta\right),$$

which grows linearly with $\epsilon$, yielding:
$$\left\|\widehat{\mathbf{A}}_\theta - \mathbf{A_\theta}\right\|_F \le \frac{L}{\mu}\epsilon.$$

$\square$

## E  PROOF OF THEOREM 3

Let $\lambda_d$ be the dominant eigenvalue of $\mathbf{A_\theta}$ with eigengap
$$\delta_\lambda = \min_{i \ne d}|\lambda_d - \lambda_i| > 0.$$

Applying the Davis–Kahan theorem for left eigenvectors, we have (Yu et al., 2015)
$$\sin\angle(\widehat{\mathbf{w}}_d, \mathbf{w}_d) \le \frac{\left\|\widehat{\mathbf{A}}_\theta - \mathbf{A_\theta}\right\|_2}{\delta_\lambda}.$$

Since
$$\|\cdot\|_2 \le \|\cdot\|_F,$$
we can further bound
$$\|\widehat{\mathbf{w}}_d - \mathbf{w}_d\|_2 \le 2\sin\angle(\widehat{\mathbf{w}}_d, \mathbf{w}_d) \le \frac{2\|\widehat{\mathbf{A}}_\theta - \mathbf{A_\theta}\|_F}{\delta_\lambda} \le \frac{2L}{\mu\,\delta_\lambda}\epsilon.$$

Hence the perturbation of the left dominant eigenvector is explicitly controlled in norm.

## F  SIMULATION MODEL DETAILS

Here, we provide a complete description of the gene-protein regulatory network model used to generate the synthetic data for our experiments.

### F.1  NETWORK STRUCTURE AND TOPOLOGY

The network consists of $N = 10$ state variables: the concentrations of 5 mRNAs $(m_0, \ldots, m_4)$ and 5 proteins $(p_0, \ldots, p_4)$. The network employs a "master regulator" topology, as depicted in Figure 1. In this structure, a single protein, $P_4$, acts as a global transcriptional activator for several genes. The concentration of $P_4$ serves as a proxy for the overall state of the system, and its self-regulation creates a positive feedback loop that gives rise to bistability and bifurcation phenomena.

## F.2 SYSTEM DYNAMICS AND GOVERNING EQUATIONS

The temporal evolution of mRNA ($m_i$) and protein ($p_i$) concentrations for each gene $i \in 0, \dots, 4$ is described by the following system of stochastic differential equations (SDEs):

$$\frac{dm_i}{dt} = \underbrace{\alpha_{\text{basal},i}}_{\text{Basal Rate}} + \underbrace{\alpha_{\text{activated}} \frac{p_4^{n_{\text{master}}}}{K_{\text{master}}^{n_{\text{master}}} + p_4^{n_{\text{master}}}}}_{\text{Activated Transcription}} - \underbrace{\gamma_{m,i} m_i}_{\text{Degradation}} + \xi_{m,i}(t)$$

$$\frac{dp_i}{dt} = \underbrace{\beta_i m_i}_{\text{Translation}} - \underbrace{\gamma_{p,i} p_i}_{\text{Degradation}} + \xi_{p,i}(t)$$

Where:

- The activated transcription term is a standard Hill function representing the cooperative binding of the master regulator $P_4$ to the promoter regions of the genes.

- $K_{\text{master}}$ is the Michaelis-Menten constant, representing the concentration of $P_4$ required for half-maximal activation. This parameter is systematically varied to induce the bifurcation.

- $n_{\text{master}}$ is the Hill coefficient, representing the cooperativity of binding.

- $\xi_{m,i}(t)$ and $\xi_{p,i}(t)$ are stochastic terms representing noise.

## F.3 NOISE MODEL

The stochastic terms $\xi(t)$ represent additive Gaussian white noise, which models random fluctuations in the biochemical reactions. The SDEs are of the form $dX_t = f(X_t)dt + \sigma dW_t$, where $f(X_t)$ is the deterministic drift part of the equations above, $dW_t$ is a Wiener process, and $\sigma$ is the noise strength.

For the numerical simulations, we use the Euler-Maruyama method with a time step $\Delta t$. The noise term for each state variable at each time step is implemented as:

$$\text{Noise} = \sigma \sqrt{\Delta t} \cdot \mathcal{N}(0, 1)$$

where $\mathcal{N}(0, 1)$ is a random variable drawn from a standard normal distribution. The noise strength $\sigma$ was set to a constant value of 0.1 for all simulations.

Snapshots are generated near the tipping point on the high branch. For each selected parameter value, we simulate 100 independent realizations (representing 100 cells) using Euler-Maruyama integration. The state of each realization at the equilibrium point is taken as a single-cell measurement, yielding a snapshot of $N = 100$ cells. This yields high-dimensional low-sample-size (HDLSS) data, which is repeated for a set of parameter values selected mainly near the bifurcation point to generate multiple snapshot datasets, with gradual shifts in distributions to test covariance estimation and re-stabilization. Ground-truth Jacobians are saved for comparison with the estimated $A_\theta$.

## F.4 PARAMETER SETTINGS

The specific parameter values used for the "Global Bifurcation" model are detailed in the table below. These values were chosen to ensure the system exhibits a clear bistable region and a saddle-node bifurcation as $K_{\text{master}}$ is varied. The bifurcation parameter $K_{\text{master}}$ is varied across the range $[0.1, 10.0]$, with denser sampling applied in the critical regions near $[0.233, 8.286]$ to better capture variance peaks from critical slowing down.

| Parameter | Value(s) | Description |
|---|---|---|
| $\alpha_{\text{basal}}$ | [0.03, 0.04, 0.06, 0.05, 0.05] | Basal transcription rate for each gene. |
| $\alpha_{\text{activated}}$ | 6.0 | Maximum activated transcription rate. |
| $K_{\text{master}}$ | Varied (e.g., logspace(-1, 1)) | Bifurcation parameter: activation constant for $P_4$. |
| $n_{\text{master}}$ | 4 | Hill coefficient for $P_4$ activation. |
| $\beta$ | [1.6, 1.9, 2.2, 2.0, 2.4] | Translation rate for each mRNA. |
| $\gamma_m$ | 1.0 (for all $i$) | mRNA degradation rate. |
| $\gamma_p$ | [1.0, 0.9, 0.8, 1.1, 1.0] | Protein degradation rate. |
| $\sigma$ (noise strength) | 0.1 | Strength of the additive Gaussian noise. |

Table 1: Parameter settings for the global bifurcation model.

## G    Limitations

While our framework provides a principled way to identify mRNA-protein regulatory dynamics from snapshot data and design re-stabilization strategies, several limitations remain. First, although the kernel conditional density estimator mitigates the high-dimension low-sample-size (HDLSS) challenge, it introduces additional computational cost and may still be sensitive to bandwidth selection. Second, the current work does not include experimental validation of the re-stabilization design. While accurate identification of regulatory nodes strongly suggests effective interventions, future studies will be needed to validate these strategies in wet-lab or clinical settings.

## H    Broader Impact

This work contributes to the emerging intersection of machine learning, biology, and medicine by providing a data-driven framework for identifying regulatory dynamics and designing early interventions in disease progression. From a biological perspective, the ability to detect pre-disease stages and suggest re-stabilization strategies has the potential to inform ultra-early treatment, shifting medical practice from reactive treatment to preventive intervention. This aligns with ongoing efforts in precision medicine, where computational tools guide targeted therapies at the molecular level. From a machine learning perspective, our study highlights how structural priors and dynamical systems theory can enhance learning in the high-dimension low-sample-size (HDLSS) regime, which frequently arises in single-cell analysis and other scientific domains. These insights may inspire future work on combining domain knowledge with statistical learning for better data efficiency and interpretability. At the same time, caution is required when interpreting computationally identified regulatory nodes as clinical intervention targets. Translational applications will require rigorous experimental validation and ethical considerations to ensure safety and effectiveness.

## I    Experiments Compute Resources

All experiments were conducted on a MacBook equipped with an Apple M4 chip and 32GB of unified memory.

## J    Reproducibility Statement

The source code has been included in the supplementary material for review purposes, and we will release it as open source if the paper is accepted, to ensure transparency and reproducibility.

