# OpenReview forum: "Learning Early Treatment Strategy from Snapshots for mRNA-Protein Regulatory Networks"
_ICLR.cc/2026/Conference — ICLR 2026 Conference Withdrawn Submission_

### Official Review · Reviewer_TW8E · 2025-10-20

**Soundness:** 2
**Presentation:** 3
**Contribution:** 2
**Rating:** 2
**Confidence:** 3

**Summary:**

The paper proposes a control-based approach to detect/ define therapeutic interventions in biological networks. It relies on an observation (itself not new) that variance of key genes increases prior to onset of disease in models where disease is considered as a bifurcation in the expression space of genes. The authors argue and show on simulations in a particular system that a linear control approach can be effective at identifying optimal interventions.

**Strengths:**

The modelling framework is generally elegant: the conceptualisation of disease as a bifurcation (not itself a contribution) opens the way to development and deployment of sound mathematical approaches to control the system (the contribution of the paper). Therefore the paper scores well on its mathematical soundness. I also appreciate the focus on a plausible practical application: while systems biology used to be very prominent in machine learning conferences a while back, this has now reduced and  bringing back some attention is worthwhile.

**Weaknesses:**

- I found the machine learning component of the work somewhat marginal, the problem is reduced to a linear control problem which is classical from the computational point of view. Perhaps the novelty from the ML perspective needs to be highlighted more?
- The modelling of cell's expression as an autonomous system in the RNA/ protein configurations space is naive and far from a realistic biological understanding, which involves multiple layers of epigenetic regulation as well as cell-to-cell communication. It might still be worthwhile, but this should be argued for and adequately qualified.
- The approach seems to focus on single-cell behaviour, but apparently ignores the substantial heterogeneity (both in terms of intrinsic but more importantly in terms of extrinsic noise such as cell types). This is likely a major issue in any actual application.
- The validation is confined to an idealised setting which is unlikely to be considered realistic by a biologist.

**Questions:**

- You mention the multi-scale nature of biological dynamics, but I don't see how it is then handled. Can you clarify?
- The proposed model seems to be autonomous, but then it appears that also parameters are time dependent (perhaps more slowly, is this the multiscale nature). Can you clarify?
- Concretely what would be the meaning of these parameters? And how could they be controlled in a real system?

---

### Official Review · Reviewer_9FQi · 2025-10-24

**Soundness:** 2
**Presentation:** 2
**Contribution:** 2
**Rating:** 4
**Confidence:** 3

**Summary:**

The authors propose an algorithm to modify the drift matrix of a linearized SDE to have smaller real parts in the eigenvalues and therefore be more stable, and apply this algorithm to simulated RNA expression and protein data.

**Strengths:**

The setup is very natural, and leads to a tractable objective given the SDE distributed data.  The goal of intervening to keep the system stable is a novel task.

**Weaknesses:**

There are some issues with the mathematical setup of the problem.  I’m somewhat puzzled by line 292 saying “measurements are collected at steady state”, since if that’s the case then the time conditioning is irrelevant and one could just use the sample covariance directly? The notation seems somewhat incoherent; on lines 295-296 it is unclear if M or N is the number of original samples or generated samples from the KDE estimate.  The same inconsistency is present between the statement of Theorem 2 and the proof: in the proof N -> \infty, should that be understood as an infinite number of samples?  In which case the KDE estimate will be perfect and the covariance bound is standard.

The main issue is the evaluation, which is only done on very small graphs with simple dynamics.  This seems not sufficiently difficult a task to validate the claim that one can learn an accurate covariance matrix in high dimensions from few samples.

A smaller issue, but the discussion of the identifiability of the system matrix A_theta is based purely on counting dimensions, which probably works with probability one given some distributional assumptions but could be degenerate for very specific choices of the true dynamics.

**Questions:**

Could the authors explain more directly 1) the role of N vs M in the notation and assumed number of samples and 2) what distribution the samples are assumed to be drawn from?

---

### Official Review · Reviewer_qdFi · 2025-10-25

**Soundness:** 2
**Presentation:** 3
**Contribution:** 2
**Rating:** 2
**Confidence:** 3

**Summary:**

The authors present a framework for identifying the dynamics of mRNA-protein regulatory networks form snapshot data. They model disease progression as involving an abrupt transition (bifurcation) from a healthy to a disease state, focusing on the unstable "pre-disease" stage before this shift. To identify the system, they estimate the state covariance matrix from the data and use it to find the system matrix. Furthermore, based on this system matrix, the authors present a re-stabilization intervention strategy which aims to make the system more stable near the bifurcation point, with the goal of preventing the system from entering the disease state.

**Strengths:**

1. The paper tackles an important problem of designing early medical interventions which is often more effective than treating late-stage disease.
2. The work proposes a theoretically grounded method to identify system dynamics from snapshot data, and establishes finite-sample confidence bounds on the accuracy of the system matrix and eigenvector estimations.

**Weaknesses:**

1. A key limitation of the study is the reliance on synthetic data for validation. Demonstrating that the method can identify known regulatory drivers or predict outcomes in an experimental dataset (e.g., from public single-cell studies of cell differentiation or disease progression) is necessary to prove its utility for real-world biological applications.
2. The work posits itself as suitable for high-dimensional data, but the simulation model uses only 5 genes and 5 proteins which is significantly smaller than the high-dimensional systems often encountered in practice (n > 10^4). Furthermore, the chosen network topology (a single protein positively affecting all genes) further limits the conclusions that can be derived at this stage as real regulatory networks involve very complex interactions that may be combinatorial in nature and may involve feedback loops. Assessing scalability, by increasing n and studying more complex networks (e.g., from STRING [1]) is crucial for understanding the method's feasibility on realistic, large-scale genomic datasets.
3. The linearization approximation (eq 2) may not fully capture the complex dynamics of the biological system. Further experimental validation is necessary to quantify how much this linearization affects the predicted system dynamics.

[1] https://string-db.org/

**Questions:**

1. Transcription and degradation are modeled as completely independent processes (eq 1). Recent work has shown that the two are often tightly coupled, with changes to mRNA degradation affecting the rate of transcription, and vice-versa [1]. Could the authors comment on how this model simplification might affect their study? I would be interested in the authors' thoughts on how this coupling could be incorporated in future work.
2. Could the authors comment on the practical feasibility of translating the re-stabilization strategy into therapeutic interventions? For instance, if the mathematical framework suggests modulating numerous targets concurrently, this would present challenges regarding delivery, specificity, and potential interactions.

[1] https://pmc.ncbi.nlm.nih.gov/articles/PMC6871655/

---

### Official Review · Reviewer_bG19 · 2025-11-02

**Soundness:** 3
**Presentation:** 3
**Contribution:** 2
**Rating:** 4
**Confidence:** 3

**Summary:**

This paper proposes a theoretically grounded framework for learning gene regulatory dynamics and designing early-stage treatment strategies directly from snapshot (non-temporal) data. The authors combine Lyapunov-based regression for system identification with finite-sample statistical guarantees and then use the identified model to construct optimal and diagonal re-stabilization controllers for pre-disease intervention. The paper is motivated by biological systems where time-series measurements are scarce and early intervention is critical (e.g., gene–protein dynamics approaching bifurcation).

The technical formulation is elegant and well motivated. The authors convincingly derive a Lyapunov regression scheme that connects steady-state covariances to the system Jacobian, allowing estimation of linearized dynamics from population snapshots. The derivations are supported by formal proofs of consistency and finite-sample bounds on both the estimated system matrix and dominant eigenvectors, giving the method a strong theoretical foundation.

Empirically, the paper validates the framework using synthetic mRNA–protein networks modeled on biologically realistic bifurcation structures (Figure 1 on page 3 and the experiment setup in Appendix F). The results demonstrate that the proposed estimator can correctly identify key regulatory nodes under high-dimensional, low-sample conditions and outperform state-of-the-art identification baselines. The presentation is clear, figures are well explained, and the connection between estimation accuracy and intervention reliability is thoughtfully articulated.

**Strengths:**

• The Lyapunov-based regression is a novel and elegant reformulation that bypasses the need for time-series data, directly linking steady-state covariances to the system matrix.
• The paper provides finite-sample theoretical guarantees for both the estimated dynamics and the dominant eigenvectors, which is rare in system identification from biological data.
• The combination of optimal and diagonal re-stabilization control designs bridges statistical learning and intervention theory, offering actionable guidance for early treatment.
• The simulation studies (Figure 3, page 9) clearly illustrate the model’s ability to recover correct regulatory nodes and the accuracy improvement with increasing sample size.
• Writing and organization are excellent; the derivations, figures, and appendices together create a coherent narrative connecting theory, algorithm, and validation.

**Weaknesses:**

• The framework is tested only on synthetic datasets, leaving open how well it performs on real biological measurements (e.g., single-cell RNA-seq or proteomics).
• Several assumptions are idealized—notably Gaussian noise, linearized dynamics near equilibrium, and smooth parameter evolution—limiting generalizability to nonlinear or non-Gaussian settings.
• The kernel conditional density estimation step may become computationally demanding and sensitive to bandwidth selection in high-dimensional space.
• Although the method identifies key regulatory nodes, the actual intervention step is not empirically validated; biological or wet-lab confirmation is deferred to future work.
• The proofs are mathematically sound but occasionally dense; summarizing key intuition (e.g., geometric meaning of eigenvector perturbation bounds) in the main text would improve accessibility.

**Questions:**

Have you tested the Lyapunov regression approach on any real single-cell RNA-seq dataset (e.g., trajectory data from reprogramming experiments)? Even a small-scale demonstration would strengthen biological relevance.

How sensitive is the system identification to bandwidth selection in the kernel conditional density estimator? Could adaptive or cross-validated bandwidths improve robustness?

The framework assumes Gaussian additive noise. How would performance change under multiplicative or heavy-tailed noise, which are common in biological systems?

---

### Note · Authors · 2025-11-12

I have read and agree with the venue's withdrawal policy on behalf of myself and my co-authors.